# First validity testing of GluciQuizz, a French self-questionnaire evaluating carb-counting for patients with type 1 diabetes

**Sopio Tatulashvili**[1,2☯], **Bleuenn Dreves**[3,4☯], **Emmanuel Cosson**[1,2☯], **Laurent Meyer**[5‡], **Remy Morello**[6,4‡], **Michael Joubert**[3,4*☯]

**1** AP-HP, Department of Endocrinology-Diabetology-Nutrition, Avicenne Hospital, Université Sorbonne Paris Nord, CINFO, CRNH-IDF, Bobigny, France, **2** Equipe De Recherche En Epidémiologie Nutritionnelle (EREN), Université Sorbonne Paris Nord and Université Paris Cité, INSERM, INRAE, CNAM, Center of Research in Epidemiology and StatisticS (CRESS), Bobigny, France, **3** Diabetes Care Unit, CHU Caen, France, **4** UNICAEN, Caen, France, **5** Diabetes Care Unit, CHU Strasbourg, France, **6** Unité De Biostatistique Et De Recherche Clinique, CHU Caen, France

☯ These authors contributed equally to this work.
‡ LM and RM also contributed equally to this work.
* joubert-m@chu-caen.fr

## Abstract

### Objective

Carbohydrate intake remains one of the most influential factors affecting post-meal glucose levels in type 1 diabetes (T1D). To individualize and optimize patient education on carbohydrate counting, it is essential to assess their knowledge of carbohydrate counting. For this purpose, we aimed to produce and validate a questionnaire for French-speaking individuals based on the American AdultCarbQuiz questionnaire (43 items).

### Design

The translation and cross-cultural adaptation of the American questionnaire were followed by an analysis conducted by patient and diabetologist expert groups.

### Participants

190 participants living with T1D, Diabetologist expert group.

### Main outcome measure(s)

Internal consistency and reliability were verified by administering the adapted French version of the questionnaire to 190 participants living with T1D.

### Analysis

Clinical validity was verified by examining the correlation between the questionnaire score and the time in range (TIR) of these participants.

### Results

Translation and back-translation demonstrated good consistency with the original questionnaire. Typical American food items were replaced with foods commonly consumed in

**Data availability statement:** All relevant data are within the manuscript and its Supporting Information files.

**Funding:** The author(s) received no specific funding for this work.

**Competing interests:** The authors have declared that no competing interests exist.

France. All items were validated by both patient and diabetologist experts. After analyzing the responses of 190 participant, seven items were considered non-discriminatory and were therefore removed. Internal consistency, assessed by Cronbach's α, was high, with α = 0.785, 95 CI [0.738–0.826]. Finally, we produced a 36 item French questionnaire named GluciQuizz. TIR correlated with the GluciQuizz score (r = 0.3; p < 0.0001).

## Conclusions and implications

This study validates a French-language self-administered questionnaire (GluciQuizz), which evaluates different domains of carbohydrate knowledge in a 15-minute quiz for people living with T1D.

## Introduction

Type 1 diabetes (T1D) is characterized by insulin deficiency, leading to high blood glucose levels. T1D is associated with both microvascular complications [1] and cardiovascular mortality [2]. To prevent these complications [3], treatment involves the replacement of both basal and prandial insulin secretion.

Glycemic control in T1D has been optimized over the past decades through multiple daily injections (MDI) of long-acting and short-acting insulin analogs, continuous subcutaneous insulin infusion (CSII) [4,5], and continuous glucose monitoring (CGM) systems [6]. More recently, metabolic control has been further improved with the introduction of hybrid closed loop (HCL) systems [7,8], which provide superior overall control compared to other insulin delivery methods, especially at night and between meals.

Despite these technological advances, post-meal glucose control remains challenging in T1D [6]. Numerous factors can influence glucose level during the four hours following meals, including insulin dose, timing of the insulin bolus, physical activity, meal composition and glycemic index [9–11]. In particular, the quantity of consumed carbohydrates remains the strongest influencing factor [12]. In view of this, patients are offered nutritional education in carbohydrate counting (in grams), to enable them to calculate the meal-time insulin dose based on the quantity of ingested carbohydrates and their insulin sensitivity [12]. Indeed, better carbohydrate knowledge is associated with improved glycemic control when paired with good adherence to carbohydrate counting practice [13,14].

To individualize and optimize the educational process, the carbohydrate-counting knowledge of patients' needs to first be evaluated. Currently, few questionnaires exist worldwide for assessing the carbohydrate-counting knowledge of individuals living with diabetes, whether they are children and parents [15,16] or adults [17,18]. The AusPCQ [16] is an adaptation of the PedCarbQuiz [15] specifically designed for Australian children and parents. Similarly, the Singapore questionnaire [18] also uses the structure of the PedCarbQuiz [15] but has modified the meal content to suit the Asian population.

The AdultCarbQuiz produced and validated in the United States of America (USA) by Watts et al. [17], was designed for Western adults. This questionnaire, though very helpful, was not well-adapted to French-European nutritional habits. In this context, we aimed to adapt and validate a French version of the questionnaire, named "GluciQuizz".

## Materials and methods

### Development of GluciQuizz: general concept and overview

The general concept of GluciQuizz remained similar to that of the AdultCarbQuiz [17]. The AdultCarbQuiz consists of 43 items divided into six Domains: Domain 1 addresses

carbohydrate food recognition (19 items), Domain 2 focuses on carbohydrate food content (7 items), Domain 3 covers nutrition label reading (4 items), Domain 4 addresses glycemic targets (4 items), Domain 5 covers hypoglycemia prevention and treatment (5 items), and Domain 6 evaluates the carbohydrate content of meals (4 items) [17].

Table 1 outlines the different steps leading to the creation of GluciQuizz, which involved a comprehensive structured process including translation and transcultural adaptation, validation, evaluation and simplification, content assessment by patients and diabetologists, evaluation of internal validity, and, finally, correlation of the GluciQuizz score with CGM outcome.

## Steps 1 and 2: translation and cross-cultural adaptation of the questionnaire

The development of the GluciQuizz was primarily driven by the desire to reflect French rather than USA dietary habits [19]. The translation of the AdultCarbQuiz from English to French was performed by a native French speaker, and the back-translation was conducted by a native English speaker who was unaware of the original questionnaire. The research group ensured that the back-translation was consistent with the original questionnaire.

Then, with the help of a dietician, some typical American food items were replaced with food items commonly consumed in France. These changes are detailed in S1–S5 Tables.

## Steps 3 and 4: validation of the adapted French version of AdultCarbQuiz

**Face validity.** To evaluate the clarity, consistency, relevance and sufficiency of the French version of the AdultCarbQuiz, 13 patients with extensive experience in T1D were recruited. Among them, 8 had never been exposed to this questionnaire (naïve), and 5 were already familiar with it. The evaluation criteria were as follows: Clarity (is the item easy to understand?), Consistency (Is the item related to the qualitative or quantitative assessment of carbohydrates?), Relevance (Is the item important?), and Sufficiency (Is the item competent?).

All patients were first asked to complete a grid using a four-point Likert scale (1 = does not meet the criteria; 2 = meets the criteria to a low degree; 3 = moderate degree; 4 = high degree)

**Table 1. Steps for Americain AdultCarbQuiz transformation to French GluciQuizz.**

| | | Denomination | Performed by |
|---|---|---|---|
| | AdultCarbQuiz | | Whatts et al. [17] |
| Step 1 | Translation and cross-cultural adaptation | French translation | Translator |
| Step 2 | | French adaptation | Dietician/ RG |
| Step 3 | Validation | Face validation | 13 expert patients |
| Step 4 | | Content validation | 4 expert diabetologists |
| Step 5 | Evaluation and simplification | Internal consistency reliability | 190 patients with T1D |
| Step 6 | | Simplification | RG |
| Step 7 | | Evaluation of Cronbach's α coefficient | RG |
| Step 8 | Clinical validation | Correlation between GluciQuizz score and percentage of time in range. | 190 patients with T1D |

RG: research group (the authors); T1D: type 1 diabetes.

to assess whether each item appeared clear, coherent, and relevant. For sufficiency, the rating was applied to each section as a whole rather than to individual items. Responses rated at 3 and 4 were considered as satisfactory.

Secondly, the patients were invited to write a free comment to highlight whether any item seemed ambiguous or inappropriately formulated.

**Content validity.** The content validity was assessed by a panel of 4 expert diabetologists, independent of the research group, all with extensive experience in therapeutic education for T1D and insulin therapy (see Acknowledgments). The aim of the questionnaire was explained to the 4 experts who were asked to freely comment on the content and presentation of each section and item.

### Steps 5 to 7: internal consistency reliability of the questionnaire

To evaluate whether the number of items could be reduced, we explored internal consistency of the adapted French version of the AdultCarbQuiz. The questionnaire was submitted through Google forms from January 1 to December 31, 2021, to adult women and men with T1D routinely using CGM. These participants were recruited from three tertiary diabetes care centers. Subjects were informed about the study objectives and did not oppose participation in the study in accordance with MR-04 regulation under French law. Oral consent was obtained, which had been validated by ethics committee (CLERS n°1992), full review was obtained. In case of oral consent, the participants were free to complete the questionnaire submitted through Google Forms. No minors were included in this study.

The percentage of correct/incorrect answers for each item was analyzed to evaluate their discrimination power.The overall Cronbach's α coefficient and its 95% confidence interval (95CI), and each Cronbach's α coefficient were calculated if an item was removed, to assess internal consistency [20].

### Step 8: clinical validation of GluciQuizz

The percentage of Time in Range (TIR, the percentage of time à person's glucose level remains within the proposed target range: 70–180mg/dL) over the last 14 days at the time of questionnaire completion for each participant was collected. Data were analyzed only if at least 70% of the time during the 14-day period was captured.

### Statistics

Quantitative variables were described using means ± standard deviations (SD) or median (range) according to distribution. Quantitative variables were described using number (n) and percentages. Pearson correlation coefficients were calculated to explore the correlation between GluciQuizz score and percentage of TIR. IBM SPSS Statistics for Windows (Version 23.0) was used.

## Results

### Steps 1 and 2: translation and cross-cultural adaptation

The original AdultCarbQuiz and its French version are provided in S1–S5 Tables. Each file contains the items corresponding to each Domain, with S4 Table combining Domains 4 and 5 of the AdultCarbQuiz (Domain 4 of the GluciQuizz), and S5 Table presenting Domain 6 of the AdultCarbQuiz (Domain 5 of GluciQuizz).

## Steps 3 and 4: validation of the adapted French version of AdultCarbQuiz

**Face validity.** Clarity, consistency, relevance and sufficiency of the adapted version of the AdultCarbQuiz sections were rated higher than 3 out of 4 by the 13 "expert" T1D patients, including both naïve patients and those already-exposed to the questionnaire, except for consistency, which was rated 2.95 out of 4 by the already-exposed patients only. Summarized results for each domain of the adapted version of the AdultCarbQuiz are presented in Table 2 (and detailed information for each item shown in S1–S5 Tables). In addition, according to the free comments, no item was considered ambiguous or inappropriately formulated.

**Content validity.** The 4 diabetologist experts independently agreed that the adapted version of the AdultCarbQuiz was well-suited to French-European dietary habits and effectively assessed patients' carbohydrate counting abilities. They also confirmed that each item fitted well with the five postulated Domains.

## Steps 5 to 7: internal consistency reliability and simplification of the questionnaire

Table 3 shows the characteristics of the 190 individuals living with T1D who completed the questionnaire.

Nine items (eight in the Domain 1, one in Domain 4) had a percentage of correct answers higher than 95% and were thus considered non-discriminatory. Seven of these items were removed, while the two were retained in Domain 1 because the research group considered these foods to be frequently consumed, maintaining their importance in the assessment.

Considering the adapted version of the AdultCarbQuiz as a global questionnaire, internal consistency assessed by Cronbach's α was high with α = 0.785, 95 CI [0.738-0.826]. We evaluated the effect of removing each of the nine items one by one on the overall Cronbach's α: 5 items appeared inconsistent, their removal resulting in a slight increase in global Cronbach's α (S1–S5 Tables). However, this slight improvement in internal consistency was not considered

**Table 2. Face validity of the five domains of French version of the AdultCarbQuiz.**

| | | Domains | | | | |
|---|---|---|---|---|---|---|
| | | **Carbohydrate food recognition** | **Carbohydrate food content** | **Nutrition label reading** | **Glycemic targets and hypoglycemia prevention and treatment** | **Carbohydrate content of meals** |
| Clarity | Questionnaire previously-exposed patients | 3.60 [2–4] | 3.49 [1–4] | 3.60 [2–4] | 3.76 [3–4] | 3.40 [1–5] |
| | Questionnaire naïve patients | 3.78 [1–4] | 4.00 [4–4] | 3.44 [1–4] | 3.78 [4–4] | 3.91 [1–4] |
| | All patients | 3.71 | 3.80 | 3.50 | 3.77 | 3.71 |
| Consistency | Questionnaire previously-exposed patients | 2.95 [1–4] | 3.60 [2–4] | 3.80 [3–4] | 3.76 [3–4] | 3.10 [1–4] |
| | Questionnaire naïve patients | 3.33 [1–4] | 3.80 [1–4] | 3.81 [3–4] | 3.85 [2–4] | 3.97 [1–4] |
| | All patients | 3.18 | 3.73 | 3.81 | 3.81 | 3.63 |
| Relevance | Questionnaire previously-exposed patients | 3.58 [2–4] | 3.59 [2–4] | 3.60 [2–4] | 3.71 [2–4] | 3.55 [1–4] |
| | Questionnaire naïve patients | 3.52 [1–4] | 3.60 [1–4] | 3.63 [2–4] | 3.83 [2–4] | 3.90 [1–4] |
| | All patients | 3.54 | 3.59 | 3.62 | 3.79 | 3.79 |
| Sufficiency | Questionnaire previously-exposed patients | 3.40 [2–4] | 3.20 [2–4] | 3.40 [2–4] | 3.67 [2–4] | 3.40 [1–4] |
| | Questionnaire naïve patients | 3.75 [3–4] | 3.75 [2–4] | 3.38 [1–4] | 3.85 [3–4] | 3.71 [1–4] |
| | All patients | 3.62 | 3.54 | 3.39 | 3.78 | 3.62 |

Clarity, consistency, relevance and sufficiency of each item of the five Domains were evaluated on a Likert scale of 1 (not at all) to 4 (totally) by 13 patients.

Questionnaire previously exposed patients (n = 5); Questionnaire naïve patients (n = 8); All patients (n = 13).

Results are expressed as means and [min-max].

**Table 3. Characteristics of patients with type 1 diabetes who answered to the questionnaire.**

| Variable | Total n = 190 |
| --- | --- |
| Age (years) | 41.5 ± 14.8 |
| Duration of diabetes (years) | 19.7 ± 12.4 |
| Insulin administration | |
| Multiple daily injection (n) | 41 |
| Continuous subcutaneous insulin injection (n) | 85 |
| Predictive low glucose suspend (n) | 20 |
| Hybrid closed loop (n) | 44 |

All data are expressed as mean ± sd.

sufficient to justify removing items that had otherwise been validated by expert patients and physicians.

These modifications resulted in the development of GluciQuizz, a French questionnaire comprising 36 items across 5 domains (S1 File).

### Step 8: clinical validation of GluciQuizz

All patients had at least 70% CGM data captured during the last 14 days. The mean TIR was 58 ± 19%. Fig 1 shows that patient GluciQuizz score correlated with the percentage of TIR (r = 0.3; p < 0.0001).

## Discussion

This study validated the new French self-administered questionnaire, GluciQuizz, adapted from the USA self-administered questionnaire, AdultCarbQuiz [17]. GluciQuizz covers a broad spectrum of carbohydrate knowledge in a test lasting approximately 15 minutes and was evaluated in this study by people living with T1D.

The first scale specifically developed to measure patients' numeracy skills in diabetes management was the Diabetes Numeracy Test [21], which tested the ability to read nutrition labels, determine the insulin glucose ratio and calculate correction factors. In 2010 Koontz published the PedCarbQuiz, a 78 item self-administered questionnaire validated for English-speaking pediatric population, targeting both children and their parents [15]. In 2011, the AdultCarbQuiz was published as the first validated self-administered questionnaire designed to evaluate carbohydrate knowledge in an English-speaking adult population

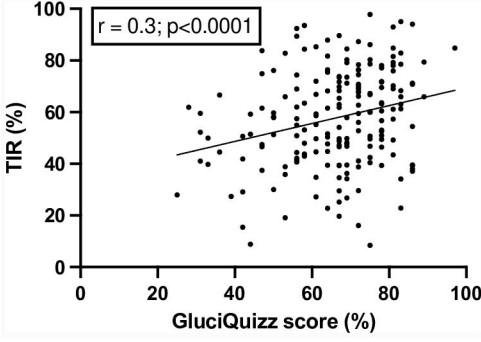

**Fig 1. Correlation between GluciQuizz score and percentage of time in range (TIR).**

[17]. Since its validation, it has been translated and adapted for different cultures. Firstly, the questionnaire was translated and adapted to include culturally-relevant foods tailored to Vietnamese women with gestational diabetes mellitus [22]. Certain items were removed, such as blackberry jam, canned spaghetti sauce, and maple syrup, while others were substituted with items more commonly consumed in Vietnam, such as sweet potatoes instead of baked potatoes and orange juice instead of apple juice. The final adapted questionnaire included 18 items. Higher scores on the questionnaire were positively-associated with self-efficacy in blood glucose management and the practice of self-monitoring blood glucose [22]. Finally, the AdultCarbQuiz has been translated into Arabic and adapted for the Saudi Arabian population by Bawazeer et al. [23]. Similar to data from the original publication of the AdultCarbQuiz [15,17,23], higher carbohydrate knowledge was associated with lower HbA1c levels in persons living with T1D [23].

Dietary measures are likely among the oldest tools for managing diabetes [24] yet carbohydrate counting remains clinically important, even for patients using the most up-to-date treatments. GluciQuizz can be used easily and rapidly by dietitians or physicians to assess their patients' carbohydrate knowledge and facilitate targeted educational reinforcement. To our knowledge, GluciQuizz is the first validated French-speaking self-administered questionnaire designed to evaluate a large spectrum of carbohydrate knowledge in just 15 minutes, with fewer questions than the AdultCarbQuiz. It should help health care professionals identify gaps in patient education and provide tailored educational program for T1D patients in order to improve their glycemic control. Indeed, better carbohydrate knowledge has been associated with improved glycemic control in T1D patients who regularly practice carbohydrate counting [13,14]. It has been shown in the study by Brazeau et al. that a mean carbohydrate counting error of 15 g per meal over three days resulted in higher glycemic variability and in decreased time with glucose values between 4 and 10 mmol/L [25]. Questionnaires that evaluate carbohydrate knowledge still appear to be crucial for patients using HCL systems. Indeed, without knowing the carbohydrate count of meals, post prandial glycemic control is suboptimal, with blood glucose levels 4 hours after meals often ranging between 170 and 200 mg/dL [26,27].

Some people living with T2D, treated by MDI or CSII, also use carbohydrate counting. Better carbohydrate knowledge was associated with lower HbA1c levels in this population in the study of Watts et al [17]. In our study, GluciQuizz was validated only in T1D patients. A future validation study specifically targeting T2D patients would be of interest.

The main strength of our study is the step-by-step adaptation and validation of the questionnaire, as recommended for cross cultural adaptation [19]. In addition, as with other adaptations of this questionnaire, we found a correlation between the questionnaire score and metabolic control. However, for the first time, our study report a CGM metric, specifically TIR[15,17,23]. We acknowledge that the use of post-prandial TIR would have been even more meaningful, but we did not have precise meal times to assess this parameter. The validation of GluciQuizz in a large panel of T1D patients on different treatments reinforces its applicability to a broader T1D population.

However, we should keep in mind that carbohydrate counting knowledge is not the only parameter determining prandial behavior in T1D patients. In addition to knowledge, regular adherence to carbohydrate counting is a key element and is associated with better glycemic control in T1D patients treated with MDI or CSII [28]. In this study, we did not assess our patients' adherence to carbohydrate counting, nor did the authors of the AdultCarbQuiz and PedCarbQuiz studies [15,17]. As patients with higher quiz scores presented better glycemic control, we can hypothesize that most participants had good adherence. A more detailed study

should now be conducted to better understand the differential implications of the level of knowledge but also the application of this knowledge in everyday life.

In conclusion, this study validates the new French-speaking self-administered questionnaire, GluciQuizz, which assesses various domains of carbohydrate knowledge in T1D patients. GluciQuizz is already being used in routine practice at several French diabetes centers and is being incorporated into clinical research programs.

## Supporting information

**S1 Table. Domain 1 of AdultCarbQuiz and GluciQuizz: carbohydrate food recognition.** The table shows the different steps in the validation process of the questionnaire (*step number as described in Table 1 of the manuscript). ACQ US, AdultCarbQuiz original version; ACQ French, AdultCarbQuiz translated into French; ACQ French adapted, questionnaire after cross-cultural adaptation for French people; Clarity, Consistency, Relevance and Sufficiency, expert notes for each item; % of correct answers of 190 participants living with T1D; Removed items, item was removed when more than 95% of patients had the same score for the same modality; Cronbach's α, inconsistent items where removal resulted in a slight increase of global Cronbach's α coefficient are presented in **bold**. * For sufficiency, the rating was applied to each section as a whole, not to individual items.
(DOCX)

**S2 Table. Domain 2 of AdultCarbQuiz and GluciQuizz: carbohydrate food content.** ACQ US, AdultCarbQuiz original version; ACQ French, AdultCarbQuiz translated into French; ACQ French adapted, questionnaire after cross-cultural adaptation for French people; Clarity, Consistency, Relevance and Sufficiency, expert notes for each item; % of correct answers of 190 participants living with T1D; Removed items, item was removed when more than 95% of patients had the same score for the same modality; Cronbach's α, inconsistent items where removal resulted in a slight increase of global Cronbach's α coefficient are presented in **bold**. * For sufficiency, the rating was applied to each section as a whole, not to individual items.
(DOCX)

**S3 Table. Domain 3 of GluciQuizz: nutrition label reading.** ACQ US, AdultCarbQuiz original version; ACQ French, AdultCarbQuiz translated into French; ACQ French adapted, questionnaire after cross-cultural adaptation for French people; Clarity, Consistency, Relevance and Sufficiency, expert notes for each item; % of correct answers of 190 participants living with T1D; Removed items, item was removed when more than 95% of patients had the same score for the same modality; Cronbach's α, inconsistent items which removal resulted in a slight increase of global Cronbach's α coefficient are presented in **bold**. * For sufficiency, the rating was applied to each section as a whole, not to individual items.
(DOCX)

**S4 Table. Domain 4 of GluciQuizz (corresponding to Domains 4 and 5 of AdultCarbQuiz): glycemic targets. hypoglycemia prevention and treatment.** ACQ US, AdultCarbQuiz original version; ACQ French, AdultCarbQuiz translated into French; ACQ French adapted, questionnaire after cross-cultural adaptation for French people; Clarity, Consistency, Relevance and Sufficiency, expert notes for each item; % of correct answers of 190 participants living with T1D; Removed items, item was removed when more than 95% of patients had the same score for the same modality; Cronbach's α, inconsistent items which removal resulted in a slight increase of global Cronbach's α coefficient are presented in **bold**. * For sufficiency, the rating was applied to each section as a whole, not to individual items.
(DOCX)

**S5 Table. Domain 5 of GluciQuizz (corresponding to Domain 6 of AdultCarbQuiz): carbohydrate content of meals.** ACQ US, AdultCarbQuiz original version; ACQ French, AdultCarbQuiz translated into French; ACQ French adapted, questionnaire after cross-cultural adaptation for French people; Clarity, Consistency, Relevance and Sufficiency, expert notes for each item; % of correct answers of 190 participants living with T1D; Removed items, item was removed when more than 95% of patients had the same score for the same modality; Cronbach's α, inconsistent items which removal resulted in a slight increase of global Cronbach's α coefficient are presented in **bold**. * For sufficiency, the rating was applied to each section as a whole, not to individual items.
(DOCX)

**S1 File. GluciQuizz.** Full questionnaire.
(DOCX)

## Acknowledgments

The authors would like to thank Dr J Morera, Pr JP Riveline, Dr A Sola-Gazagnes and Dr A Rod for their expert review of the GluciQuizz.

The authors also thank S Watts, J Anselmo and E Kern for permission to adapt the AdultCarbQuiz for French-speakers.

Back-translation of GluciQuizz and editorial assistance in the preparation of this paper was provided by Dr Ian Darby. This editorial assistance was funded by the authors.

## Author contributions

**Conceptualization:** Remy Morello, Michael Joubert.

**Data curation:** Sopio Tatulashvili, Emmanuel Cosson, Laurent Meyer, Michael Joubert.

**Formal analysis:** Remy Morello.

**Methodology:** Bleuenn Dreves, Michael Joubert.

**Validation:** Bleuenn Dreves, Emmanuel Cosson, Laurent Meyer, Remy Morello, Michael Joubert.

**Visualization:** Michael Joubert.

**Writing – original draft:** Sopio Tatulashvili, Emmanuel Cosson, Michael Joubert.

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
