## [Decision Letter · Decision Letter 0]

16 Jul 2024

PONE-D-24-21453First validity testing of GluciQuizz, a French self-questionnaire evaluating carb-counting for patients with type 1 diabetesPLOS ONE

Dear Dr. Joubert,

Thank you for submitting your manuscript to PLOS ONE. After careful consideration, we feel that it has merit but does not fully meet PLOS ONE’s publication criteria as it currently stands. Therefore, we invite you to submit a revised version of the manuscript that addresses the points raised during the review process. You can find below the reviewers comments that must be addressed before reconsidering pubblication.

We look forward to receiving your revised manuscript.

Kind regards,

Andrea Da Porto

Academic Editor

PLOS ONE

Journal Requirements:

3. Please review your reference list to ensure that it is complete and correct. If you have cited papers that have been retracted, please include the rationale for doing so in the manuscript text, or remove these references and replace them with relevant current references. Any changes to the reference list should be mentioned in the rebuttal letter that accompanies your revised manuscript. If you need to cite a retracted article, indicate the article’s retracted status in the References list and also include a citation and full reference for the retraction

Reviewers' comments:

Reviewer's Responses to Questions

**Comments to the Author**

1. Is the manuscript technically sound, and do the data support the conclusions?

Reviewer #1: Yes

Reviewer #2: Yes

Reviewer #3: Yes

2. Has the statistical analysis been performed appropriately and rigorously? 

Reviewer #1: Yes

Reviewer #2: I Don't Know

Reviewer #3: Yes

3. Have the authors made all data underlying the findings in their manuscript fully available?

Reviewer #1: No

Reviewer #2: Yes

Reviewer #3: Yes

4. Is the manuscript presented in an intelligible fashion and written in standard English?

Reviewer #1: Yes

Reviewer #2: Yes

Reviewer #3: No

5. Review Comments to the Author

Reviewer #1: I reviewed the article untitled "First validity testing of GluciQuizz, a French self-questionnaire evaluating carb-counting for patients with type 1 diabetes" which addresses the need for a French questionnaire to assess carbohydrate counting skills in French adults with T1D.

The cultural adaptation of such a questionnaire is a thorough work that requires several stages and participants. Overall, the manuscript is well written, and each step is well described. Therefore, most of my comments are minor.

A small note, I was not able to access the supplementary material, therefore some of my comments might have been addressed if I had access to it.

1. In the introduction, line 60, the authors state that HCL allows optimal glycemic control at night and between meals. I would rephrase this to say that HCL provides better overall glycemic control (compared to other modes of insulin delivery), especially control at night and between meals.

2. In the introduction, line 64, the authors refer to meals and carbohydrate quality. Do they mean meal composition and carbohydrate counting (according to the reference)?

3. In the methods (page 7, first paragraph), the cut-off to decide whether clarity/consistency/relevance... is acceptable is not defined a priori.

4. In Table 1, last row, some text seems to be missing "Correlation between GluciQuizz" and?

5. On page 8, step 8, one could say that assessing post-meal TIR would have been even more sensitive. Just a comment.

6. On page 8, line 150, I do not clearly understand why there are only 5 domains in the French version versus 6 domains in the US version. Were two domains combined and why?

7. In Table 2, it would have been more readable to have the full domain title in the heading instead of the numbers. Same for AE/N/A. I think this could be done by changing the table to a landscape layout and putting the range on the same row as the mean, thus reducing the height of the rows.

8. In Table 3, it would have been nice to show the educational level of the patients who answered the questionnaire, since carbohydrate counting skills are associated with socioeconomic and education background. Especially since the authors found 9 items to be non-discriminatory due to a high rate of correct answers.

9. On page 13, line 187-188, sorry if this sounds silly as I am not a questionnaire expert, but would it make sense to calculate the Cronbach separately for each group of domains assessed? Or is only the overall Cronbach important?

Reviewer #2: Thank you for inviting me to review this manuscript “First validity testing of GluciQuizz, a French self-questionnaire evaluating carb-counting for patients with type 1 diabetes.

The article is presented in simple language and is easy to follow. The authors have done a comprehensive validity testing process including, translation, adaptation, back translation, testing face validity, testing with target group and experts for the content and reliability testing in a systematic way.

Reliability coefficients of 0.738-0.826 indicates high internal consistency. I believe predictive validity is not required as it is an adapted version for lanuage and minor cultural preferences.

I am not proficient in French and cannot comment for any language error in the Gluciquizz.

Reviewer #3: Dear Authors,

This French-speaking self-administered GluciQuizz validation study questionnaire is essential to French patients. Studies show that carbohydrate counting is a valuable tool for better glucose control and reduced hypoglycemia, and it is crucial to evaluate carbohydrate knowledge and cross-cultural adaptation to help patients learn the carbohydrate count appropriately. The authors followed all the steps recommended for a validation study. However, the manuscript deserves writing revision due to some grammar, punctuation, and spelling mistakes.

6. PLOS authors have the option to publish the peer review history of their article (what does this mean? ). If published, this will include your full peer review and any attached files.

**Do you want your identity to be public for this peer review?** For information about this choice, including consent withdrawal, please see our Privacy Policy .

Reviewer #1: No

Reviewer #2: No

Reviewer #3: No

---

## [Author Response · Author response to Decision Letter 0]

15 Jan 2025

5. Review Comments to the Author

Reviewer #1: I reviewed the article untitled "First validity testing of GluciQuizz, a French self-questionnaire evaluating carb-counting for patients with type 1 diabetes" which addresses the need for a French questionnaire to assess carbohydrate counting skills in French adults with T1D.

The cultural adaptation of such a questionnaire is a thorough work that requires several stages and participants. Overall, the manuscript is well written, and each step is well described. Therefore, most of my comments are minor.

A small note, I was not able to access the supplementary material, therefore some of my comments might have been addressed if I had access to it.

Response: Thank you for your positive evaluation of our work.

1. In the introduction, line 60, the authors state that HCL allows optimal glycemic control at night and between meals. I would rephrase this to say that HCL provides better overall glycemic control (compared to other modes of insulin delivery), especially control at night and between meals.

Response: Thank you, we changed the sentence.

2. In the introduction, line 64, the authors refer to meals and carbohydrate quality. Do they mean meal composition and carbohydrate counting (according to the reference)?

Response: Indeed, we have replaced the word ‘quality’ by ‘meal composition and glycemic index’ in the manuscript.

3. In the methods (page 7, first paragraph), the cut-off to decide whether clarity/consistency/relevance... is acceptable is not defined a priori.

Answer: We have added the sentence to the manuscript: ‘responses rated at 3 and 4 were considered as satisfactory’.

4. In Table 1, last row, some text seems to be missing "Correlation between GluciQuizz" and?

Answer: thank you for this remark. We have completed the table 1 with the following sentence: ‘Correlation between GluciQuizz score and percentage of time in range (TIR)’.

5. On page 8, step 8, one could say that assessing post-meal TIR would have been even more sensitive. Just a comment.

Answer: We agree with your comment; however, unfortunately, we did not have information about precise meal times to assess post-meal TIR. We added this comment in the discussion.

6. On page 8, line 150, I do not clearly understand why there are only 5 domains in the French version versus 6 domains in the US version. Were two domains combined and why?

Answer: Indeed, domains 4 and 5 of the original questionnaire were combined, as they explore similar areas, such as glycemic targets and the use of carbohydrates in various situations like hypoglycemia and physical activity.

7. In Table 2, it would have been more readable to have the full domain title in the heading instead of the numbers. Same for AE/N/A. I think this could be done by changing the table to a landscape layout and putting the range on the same row as the mean, thus reducing the height of the rows.

Answer: We have made the changes to table 2 as suggested

8. In Table 3, it would have been nice to show the educational level of the patients who answered the questionnaire, since carbohydrate counting skills are associated with socioeconomic and education background. Especially since the authors found 9 items to be non-discriminatory due to a high rate of correct answers.

Answer: We agree with this comment; however, we unfortunately do not have information about the education level of participants. Another study, which is currently in progress, is addressing this question.

9. On page 13, line 187-188, sorry if this sounds silly as I am not a questionnaire expert, but would it make sense to calculate the Cronbach separately for each group of domains assessed? Or is only the overall Cronbach important?

Answer: It is indeed an excellent question that would make sense if the scale could be used only for one of its domains or another. This scale is interesting in a global approach, hence our choice to validate it in its entirety.

Comment: Thank you very much for your careful review and your constructive remarks.  

Reviewer #2: Thank you for inviting me to review this manuscript “First validity testing of GluciQuizz, a French self-questionnaire evaluating carb-counting for patients with type 1 diabetes.

The article is presented in simple language and is easy to follow. The authors have done a comprehensive validity testing process including, translation, adaptation, back translation, testing face validity, testing with target group and experts for the content and reliability testing in a systematic way.

Reliability coefficients of 0.738-0.826 indicates high internal consistency. I believe predictive validity is not required as it is an adapted version for lanuage and minor cultural preferences.

I am not proficient in French and cannot comment for any language error in the Gluciquizz.

Comment: Thank you very much for your careful review and your constructive remarks.  

Reviewer #3: Dear Authors,

This French-speaking self-administered GluciQuizz validation study questionnaire is essential to French patients. Studies show that carbohydrate counting is a valuable tool for better glucose control and reduced hypoglycemia, and it is crucial to evaluate carbohydrate knowledge and cross-cultural adaptation to help patients learn the carbohydrate count appropriately. The authors followed all the steps recommended for a validation study. However, the manuscript deserves writing revision due to some grammar, punctuation, and spelling mistakes.

Answer: Thank you very much for your positive evaluation of our work. We have taken this remark into account and made the necessary corrections to the manuscript.

---

## [Editor Report · Decision Letter 1]

21 Jan 2025

First validity testing of GluciQuizz, a French self-questionnaire evaluating carb-counting for patients with type 1 diabetes

PONE-D-24-21453R1

Dear Dr. Michael Joubert

We’re pleased to inform you that your manuscript has been judged scientifically suitable for publication and will be formally accepted for publication once it meets all outstanding technical requirements.

Kind regards,

Andrea Da Porto

Academic Editor

PLOS ONE

---

## [Editor Report · Acceptance letter]

PONE-D-24-21453R1

PLOS ONE

Dear Dr. Joubert,

I'm pleased to inform you that your manuscript has been deemed suitable for publication in PLOS ONE. Congratulations! Your manuscript is now being handed over to our production team.

Kind regards,

on behalf of

Dr. Andrea Da Porto

Academic Editor

PLOS ONE